# APRF1 Interactome Reveals HSP90 as a New Player in the Complex That Epigenetically Regulates Flowering Time in *Arabidopsis thaliana*

**DOI:** 10.3390/ijms25021313

**Published:** 2024-01-21

**Authors:** Ioannis Isaioglou, Varvara Podia, Athanassios D. Velentzas, Georgios Kapolas, Despoina Beris, Michael Karampelias, Panagiota Konstantinia Plitsi, Dimitris Chatzopoulos, Despina Samakovli, Andreas Roussis, Jasmeen Merzaban, Dimitra Milioni, Dimitrios J. Stravopodis, Kosmas Haralampidis

**Affiliations:** 1Section of Botany, Biology Department, National and Kapodistrian University of Athens, 15772 Athens, Greece; ioannis.isaioglou@kaust.edu.sa (I.I.); vapodia@biol.uoa.gr (V.P.); gkapolas@yahoo.gr (G.K.); d.mperi@bpi.gr (D.B.); dsamakovli@biol.uoa.gr (D.S.); aroussis@biol.uoa.gr (A.R.); 2Bioscience Program, Biological and Environmental Science and Engineering Division, King Abdullah University of Science and Technology (KAUST), Thuwal 23955, Saudi Arabia; mike.karampelias@gmail.com (M.K.); jasmeen.merzaban@kaust.edu.sa (J.M.); 3Section of Cell Biology & Biophysics, Biology Department, National and Kapodistrian University of Athens, 15772 Athens, Greece; tveletz@biol.uoa.gr (A.D.V.); dimitrischat@gmail.com (D.C.); dstravop@biol.uoa.gr (D.J.S.); 4Department of Agricultural Biotechnology, Agricultural University of Athens, Iera Odos 75, 11855 Athens, Greecedmilioni@aua.gr (D.M.)

**Keywords:** Arabidopsis, flowering, WD40, WDRs, Co-IP, APRF1, HSP90

## Abstract

WD40 repeat proteins (WDRs) are present in all eukaryotes and include members that are implicated in numerous cellular activities. They act as scaffold proteins and thus as molecular “hubs” for protein–protein interactions, which mediate the assembly of multifunctional complexes that regulate key developmental processes in *Arabidopsis thaliana*, such as flowering time, hormonal signaling, and stress responses. Despite their importance, many aspects of their putative functions have not been elucidated yet. Here, we show that the late-flowering phenotype of the *anthesis promoting factor 1* (*aprf1*) mutants is temperature-dependent and can be suppressed when plants are grown under mild heat stress conditions. To gain further insight into the mechanism of APRF1 function, we employed a co-immunoprecipitation (Co-IP) approach to identify its interaction partners. We provide the first interactome of APRF1, which includes proteins that are localized in several subcellular compartments and are implicated in diverse cellular functions. The dual nucleocytoplasmic localization of ARRF1, which was validated through the interaction of APRF1 with HEAT SHOCK PROTEIN 1 (HSP90.1) in the nucleus and with HSP90.2 in the cytoplasm, indicates a dynamic and versatile involvement of APRF1 in multiple biological processes. The specific interaction of APRF1 with the chaperon HSP90.1 in the nucleus expands our knowledge regarding the epigenetic regulation of flowering time in *A. thaliana* and further suggests the existence of a delicate thermoregulated mechanism during anthesis.

## 1. Introduction

Living systems exhibit complex organization through functional networks of regulatory interactions, with scaffolding proteins being key components in controlling molecular interactions inside cells. Studies have shown that WD40 repeat proteins (WDRs) act as scaffolding molecules, assisting the reversible assembly of complexes, the proper activity of other functional proteins, and subsequent protein interactions [1,2]. WDRs are characterized by a β propeller structure made up of 4–16 WD40 motifs, which comprise one or more WD40 interacting domains, where interaction with other proteins occurs [3]. WDRs are prevalent in eukaryotes and are involved in various cellular, metabolic, and molecular pathways, suggesting an ancient origin [4].

In yeast, the WDR Swd2 is vital for survival and shows subfunctionalization as it participates in two protein complexes. As a subunit of the COMPASS (Complex Proteins Associated with Set1) complex, it contributes to histone H3 lysine 4 methylation, and as part of the CPF (Cleavage and Polyadenylation Factor) complex, it aids in RNA polymerase II transcription termination and 3′ end processing [5,6,7]. In plants, WDRs also play important biological roles in various developmental processes and stress signaling. In *A. thaliana*, the WD40 protein TWD40-2 is involved in the regulation of cellulose synthesis and endocytosis, while the mutation of the TWD40-2 gene results in impaired endocytosis, the unregulated over-accumulation of cellulose synthase A (CESA) at the plasma membrane, decreased cellulose content, and stunted plant growth [8]. Some WD40 proteins have been shown to function as substrate receptors in Cullin 4 RING-dependent E3 ubiquitin ligase-mediated proteasomal degradation and DNA damage repair mechanisms [9]. DDB1-CUL4 ASSOCIATED FACTOR (DCAF1), for example, has been shown to form a distinct Arabidopsis E3 ubiquitin ligase complex that regulates multiple developmental processes. Homozygous *dcaf1* mutants exhibit several developmental defects, including smaller seedlings with abnormal phyllotaxy, multiple primary shoots, and irregular branching [10]. Furthermore, the small WDR VERNALIZATION INDEPENDENT 3 (VIP3) has been shown to associate with DAMAGED DNA BINDING PROTEIN 1 (DDB1) [11]. Together, they are subunits of a putative E3 DWD CRL complex that negatively regulates flowering time in *A. thaliana* through the ubiquitin-mediated degradation of specific target proteins [11,12]. Mutations in the *VIP3* gene result in pleiotropic phenotypes, such as stunted growth, small rosette leaves, and flower abnormalities. In monocotyledonous wheat, TaWD40D encodes a member of the WD40 family of proteins and functions as a positive regulator of plant responses to salt and osmotic stress [13], while in rice, the protein encoded by LOC_Os08g38880 has five WD40 repeats and the corresponding gene is regulated by salt stress [14]. On the other hand, Arabidopsis TRANSPARENT TESTA GLABRA1 (TTG1) is involved in the regulation of anthocyanin biosynthesis, which is induced by various abiotic stresses such as drought, high salinity, and UV-B radiation [9]. It has been reported that anthocyanin biosynthesis is also regulated by heat shock factors and proteins (HSPs), which are induced by heat stress [15].

We have previously shown that the Arabidopsis genome contains two closely related low-molecular-weight WDR homologs (ULCS1 and APRF1) with distinct functions. ULCS1 is a subunit of an E3 Cullin Ring Ligase complex that regulates secondary wall modification proteins [16]. The deregulation (RNAi silencing or over-expression) of *ULCS1* results in pleiotropic developmental phenotypes, including a reduced deposition of secondary cell wall components and lignin in the vascular system, sterility, variable seed size, accelerated growth, and enlarged flower organs [16,17]. APRF1, on the other hand, displays functional homology to the Swd2 subunit protein of the yeast histone methylation COMPASS complex [5,6]. The total loss-of-function *aprf1* mutants display a prolonged vegetative growth phase, resulting in a delayed transition to flowering [18]. Here, we demonstrate that this late-flowering phenotype is temperature-dependent and can gradually be suppressed by elevated growth temperatures. Given the nature of APRF1 as a scaffold protein and its putative role in promoting thermoregulated flowering in Arabidopsis, we performed a co-immunoprecipitation (Co-IP) protocol using the in planta-expressed FLAG-tagged APRF1 protein, followed by mass-spectrometry-based proteomics. Our results showed that APRF1 interacts with numerous cytoplasmic and nuclear proteins, including chaperons that are involved in signaling pathways, RNA binding and processing, E3 ubiquitin ligase, and histone modification complexes. We further verified that APRF1 interacts specifically with HSP90.1 in the nucleus and with HSP90.2 in the cytoplasm, indicating its involvement in a putative thermoregulated flowering mechanism in *Arabidopsis thaliana*.

## 2. Results

### 2.1. APRF1-Impaired Plants Display a Temperature-Dependent Flowering Time Phenotype

By using molecular and genetic approaches, we have previously characterized the low-molecular-weight WD40 repeat (WDR) protein APRF1 from Arabidopsis as a key regulator of flowering time. APRF1 is the functional homolog of the Swd2 subunit of the yeast histone methylation COMPASS complex. The disruption of *APRF1*’s function most likely interferes with the proper epigenetic regulation of *FLOWERING LOCUS C* (*FLC*) transcript levels, leading to a late-flowering phenotype under long days (LDs) [18]. To investigate whether this late-flowering phenotype can also be reproduced under short-day environmental conditions, seed germination and plant growth were also tested under a photoperiodic regime of 8 h of light and 16 h of darkness (short days, SDs). Since under SDs, the vegetative growth phase of Arabidopsis is significantly extended and the transition to the reproductive phase varies among individuals, we grew the plants initially under SD conditions for two months (60 days) and then transferred them to LD conditions for 14 days. As shown in Figure 1a, the WT (wild-type, control) plants displayed a fully expanded phenotype with the development of multiple axillary branches on the primary inflorescent stem as well as lower immerging secondary shoots. On the contrary, a short primary stem of only a few centimeters in length with no visible secondary inflorescent shoots emerging from the rosette was observed in *aprf1* mutants. However, this primary stem displayed an increased number of axillary buds, which could further continue normal development.

To test whether the shoot apical meristems (SAMs) of the *aprf1* mutant plants display a canonical gene expression profile of key regulators that control meristem identity and function, we performed an RT-PCR expression analysis using apical tissues obtained from the wild-type (WT) and *aprf1* plants. Our results showed non-detectable *APRF1* expression in the mutants; however, the expression of all the key loci examined, such as *WUSCHEL* (*WUS*), *SHOOT MERISTEMLESS* (*STM*), and *CLAVATA 2* (*CLV2*), displayed wild-type expression transcript levels (Figure 1b). The only significant alterations in the *aprf1* SAM architecture were the increased width and cell number in the L1 layer of the apical meristem, as the *aprf1* plants displayed a slightly increased number of L1 cells in the meristem compared to the WT plants (Figure 1c).

Previous studies showed that *APRF1* loss-of-function mutants display a late-flowering phenotype when grown at 21–22 °C, either under LD or SD conditions. While WT plants flower about 26 days after sowing (DAS), *aprf1* mutants enter their reproductive phase about 7 days later and flower 32–33 DAS (Figure 1d). However, the preliminary results pointed out that when the growing temperature was adjusted to below 21–22 °C, the difference in flowering time between the WT and mutant plants increased. The observation that flowering time gradually increased in a mode related to the decrease in temperature led us to the hypothesis that the late-flowering phenotype of the *aprf1* plants could be temperature-dependent. To address this, we investigated whether an opposite effect could also be triggered by measuring the flowering time of WT and *aprf1* mutant plants grown at elevated temperatures. As shown in Figure 1e, both the WT and *aprf1* plants displayed a similar and synchronized flowering time at 26 °C. To confirm this observation statistically, we also conducted a thorough phenotypic analysis regarding the flowering time of 500 individual wild-type and *aprf1* mutants grown at 21 °C and 26 °C. Our results showed that at 21 °C, 50% of the WT plants flowered approximately 31 DAS, while in the *aprf1* mutants, the emergence of the primary shoot in 50% of the population occurred 37–38 DAS (Figure 1f). On the contrary, when the plants were grown at 26 °C, the transition to flowering was almost synchronized in the WT and *aprf1* mutants and took place approximately 28–29 DAS (Figure 1g).

### 2.2. APRF1 Protein Is Localized Both in the Nucleus and the Cytoplasm

WDRs act as scaffolding subunits in various regulatory complexes, functioning in many subcellular compartments and developmental processes, including histone methylation and protein ubiquitination [19,20]. In the *A. thaliana* proteome, more than 80 WDRs have been identified as putative E3 DWD CRL subunits, including COP1, DCAF1, MSI4, and ULCS1, the closely related homolog of APRF1 [16] (and references therein). APRF1 is a small-molecular-weight WDR protein which contains seven WD40 repeats that span almost the entire protein sequence (Figure 2a,b). The existence of so many WDRs containing solely WD40 repeats indicates that both the variability of the complexes and the range of processes being affected by them may be even greater than anticipated. We therefore decided to have a closer look into the subcellular localization of the APRF1 protein. Stable transgenic Arabidopsis lines were generated using a YFP::APRF1 translational fusion construct, and YFP expression was carefully monitored in the root tissues at various developmental stages. The results showed that in differentiated cells, like mature root cells or trichomes, APRF1 is predominantly localized in the nucleus [18]. However, in the epidermal cells of the root meristematic zone, a strong YFP signal was recognized in both the nucleus and the cytoplasm (Figure 2c). The above findings indicate that APRF1 has dual subcellular placement, participating most likely in distinct molecular or signaling mechanisms, while its localization is developmentally regulated.

### 2.3. APRF1 Co-IP Resulted in 471 Non-Redundant Putative Interacting Proteins

In a previous report, we functionally characterized two *A. thaliana* T-DNA loss-of-function lines (*aprf1-7* and *aprf1-9*). Both mutants displayed a similar late-flowering phenotype, which is likely attributed to the epigenetic deregulation of *FLC* expression. Given the functional complementation of the yeast Swd2 mutant strain (impaired in H3K4 histone methylation) by the *APRF1* plant ortholog, we proposed that APRF1 is probably a subunit of a methylation complex that controls flowering time in Arabidopsis through the epigenetic regulation of FLC expression [18]. In the present study, our results further demonstrate that this late-flowering phenotype of the *aprf1* mutant plants is temperature-dependent, suggesting that the manifestation of the phenotype may involve additional components that may stabilize the function of the FLC regulatory complex in elevated temperature conditions.

To identify these components/interactors of APRF1, we employed a co-immunoprecipitation (Co-IP) approach. We therefore generated an ARPF1-FLAG-tagged construct to specifically pulldown complexes that contain the APRF1 protein. The construct was introduced into *A. thaliana aprf1* mutant plants, and independent transgenic lines were generated. An equal amount of protein total lysate from the wild-type (WT) and the transgenic lines was blotted against an anti-FLAG antibody to verify the expression of the recombinant APRF1-FLAG protein, which has a molecular weight of approximately 37 kDa [18]. The two lines, line 1 and line 3, showing efficient expression of the AFRF1-FLAG protein were used in the downstream Co-IP experiments (Figure 3a, upper panel). As a quality control experiment for the Co-IP approach, 5 μL at every step, starting from the isolation of the total protein up to the elution of the immunoprecipitation product, was collected and analyzed by immunoblot assays. The lower panel of Figure 3a shows a representative head-to-head comparison between the line 1 plants, indicated as “1”, and the wild-type (WT) plants, indicated as “Ø”. Initially, the sample corresponding to the total lysate of line 1 did not show any band of the protein of interest in the total lysate (TL) before concentration (BC) (TL, BC). However, after concentrating the sample (AC), the band of interest had a clear signal (TL, AC). Finally, the Co-IP product from the concentrated lysate revealed a strong band in line 1, representing the specific detection of the APRF1 FLAG-tagged protein by the FLAG antibody (Figure 3a, Co-IP). At the same time, only a tiny portion of the protein of interest was found in the flow-through (FT) sample, indicating that APRF1-FLAG was successfully extracted and immunoprecipitated from the plants.

After confirming the neat pulldown of APRF1-FLAG, the co-immunoprecipitation (Co-IP) products of the line 1, line 3, and wild-type (WT) plants were prepared and run for an MS analysis. As illustrated in Figure 3b, from each sample, a few hundred proteins were detected. The stringency of the analysis was adjusted to filter out non-specific interactions and allow, at the same time, for the identification of weak and/or indirect interactions. The experiments using APRF1-FLAG as bait yielded a total of 1336 non-redundant proteins from mutant line 1, 1210 proteins from mutant line 2, and 640 non-redundant proteins when pulldown was conducted in the background of the WT Col-0 plants (control sample). Out of the 640 proteins identified in the control sample (WT), 548 were common in all three plant backgrounds. It is worth mentioning that the APRF1 protein was identified by mass spectrometry (MS) in lines 1 and 3, but not in the Col-0 plants, confirming both the efficacious pulldown and the high specificity of the Co-IP products. By using the specialized bioinformatics platform FLAME, the proteins detected only in the line 1 and line 3 samples were separated from the ones detected in the control sample (WT), considered background signals. This resulted in a list of 471 non-redundant proteins, which were valued to be the most confident true interactors and thus further used in the downstream analyses.

### 2.4. Functional Enrichment Profiling

To uncover biological insights from the herein unveiled 471 non-redundant proteins and to provide the appropriate biological context regarding their pathways of involvement, we next proceeded to analyze their functional enrichment via the FLAME bioinformatics platform [21]. Functional enrichment analyses are a widely used method for interpreting experimental results by identifying categories of functional terms in which genes or proteins have been found to be over-represented. A “Gene Ontology” (GO) and KEGG enrichment analysis were performed using the gProfiler enrichment tool, which tests for statistically significant enrichment by comparing an experimental list to a background set of organism-specific (e.g., *A. thaliana*) genes annotated in the Ensembl database [22]. The resulting *p*-values were corrected for the Benjamini–Hochberg false discovery rate (FDR). The obtained data were presented in a Manhattan plot (Figure 4a), in which the functional terms are organized along the *x*-axis and colored according to their data source, while the *y*-axis indicates the significance (*p*-value) of each term. The FLAME-mediated functional enrichment profiling of the 471 non-redundant proteins revealed their critical involvement in 172 molecular functions, 126 cellular components, 348 biological processes, and 21 KEGG pathways (Figure 4 and Appendix A). In the ontology of “Molecular Function” (MF), “binding” (including the binding of small molecules, nucleotides, etc.) was presented as the term with the lowest *p*-value (Figure 4b). Notably, this category is highly represented (81.63%) in our experimental list. Regarding cellular components, “proteolysis”, via the engagement of “proteasome” activity, was detected among the terms with the highest enrichment score, whereas the “protein-containing complex” was classified among those with the lowest *p*-value. The GO enrichment for biological processes identified several terms related to “translational initiation” and “proteolysis”. Most importantly, the KEGG pathway enrichment analysis revealed several terms, with “amino acid and secondary metabolite biosynthesis” and “proteasome” being clustered among the most significantly enriched pathways (Figure 4b).

### 2.5. Protein–Protein Interaction Analysis

To gain putative mechanistic insights into our, enriched functions and pathway data, we used the STRING platform [23] to analyze protein–protein interactions. The STRING database is a powerful tool for investigating protein–protein interactions and their roles in various biological processes. STRING supports both physical and functional interactions, with physical interactions referring to proteins that are part of the same bio-molecular complex, while functional interactions refer to proteins involved in the same pathway or biological process. In the resulting networks, the interacting entities are represented as nodes and their interactions (evidence or confidence) as edges. In confidence mode, the thickness of the edge reflects the interaction score. Thereby, we focused on studying the interactions of proteins belonging to the “protein-containing complex”, “translation”, and “RNA binding” categories (Figure 4b), in which WDRs are known to participate in the assembly of multi-functional complexes. The obtained STRING network indicated two major clusters of interacting proteins: those related to “translation” (red) and those related to “proteolysis” (blue) (Appendix A). The APRF1 protein, which contains WD40 repeats, is positioned near the cluster of translation-related proteins. Although our laboratory results suggest that the late-flowering phenotype associated with APRF1 is temperature-dependent, the STRING network analysis of proteins known to belong to the “protein folding” and “chaperone” categories showed a lack of interactions with the APRF1 protein (Appendix A), suggesting that this association remains unknown and requires revision.

### 2.6. Molecular Docking Analysis of APRF1 with HSP90.1 and HSP90.2

The functional enrichment analysis revealed that more than 80% of our 471 non-redundant proteins belong to the “GO:MF” binding category, in which APRF1 co-exists with the HSP90.2 chaperone. Therefore, in order to assess the physical interactions between APRF1 and each one of the two members of the HSP90 chaperone family, HSP90.1 and HSP90.2, molecular docking experiments were conducted using the ClusPro 2.0 protein docking server with default parameters. ClusPro is a web-based protein–protein rigid-body docking tool which provides docked structures based on the total energy functions of the produced complexes [24,25,26,27]. The ClusPro binding analysis generated 29 models for both the APRF1–HSP90.1 and APRF1–HSP90.2 heterodimeric interactions. The generated clusters were then evaluated using the balanced scoring coefficients scheme, and the models with the lowest center-weighted score were selected and visualized using PyMOL 2.6 software [28] (Figure 5). An interaction interface analysis of these complexes through PDBsum [29,30] revealed that the APRF1 and HSP90.1 interaction may occur through the formation of nine salt bridges, 22 hydrogen bonds, and 266 non-bonding contacts, while the interaction of APRF1 with HSP90.2 may occur through the formation of nine salt bridges, 23 hydrogen bonds, and 228 non-bonding contacts. The high number of distinct binding sites, all present in the docking complexes, provides clear evidence for the strong interactions between APRF1 and HSP90.1 (Figure 5a) and APRF1 and HSP90.2 (Figure 5b).

### 2.7. APRF1 and HSP90 Chaperons Functionally Interact in Planta

In view of the physiological data regarding the thermodynamic complementation of the late-flowering phenotype of the *aprf1* mutants by mild heat stress, along with the obtained Co-IP data and the results from the virtual docking predictions, we tested whether APRF1 indeed interacts in planta with members of the HSP90 family from Arabidopsis. For this, we chose HSP90.1, a heat-inducible member [31], and HSP90.2, a constitutively expressed member [32]. By performing bimolecular fluorescence complementation (BiFC) assays using the appropriate negative controls, we show that both HSP90 proteins physically interact with APRF1 in planta, resulting in a strong reconstruction of the YFP signal (Figure 6). Surprisingly, but concurrently very interesting, is the fact that the interaction of APRF1 appears to occur in different subcellular compartments, depending on the HSP90 member. Our results show that APRF1 interacts with HSP90.1 specifically in the nucleoplasm and nucleolus (Figure 6a) and with the HSP90.2 protein specifically in the cytoplasm of the cells (Figure 6b), indicating different functions of APRF1/HSP90 heterocomplexes. The co-transformation of the construct APRF1–YFPn along with the empty YFPc vector or HSP90.1/Hsp90.2 and the empty YFPn vector did not result in any fluorescence signal (Figure 6c).

## 3. Discussion

It is generally accepted that many WDRs which contain one or more WD40 motifs can act as scaffold proteins for the assembly of large molecular machines. Despite their simplicity, they play central roles in regulating various environmental stress responses or developmental processes, like seed germination and flowering, by acting as hubs for protein–protein or protein–nucleic acid interactions [33]. They can dock a variety of substrates, with similar or distinct modes, by utilizing the entire surface of their β-propeller architecture and thus participating as subunits in various multi-protein complexes. The cellular processes and networks that WDRs have been identified as key regulators of include signal transduction, cell cycle control, RNA processing, histone methylation, and protein ubiquitination [20,33]. In signaling pathways, WDRs are involved in the perception and transduction of various plant hormones, such as auxin and brassinosteroids, by mediating the interaction between hormone receptors and downstream effectors [34]. For example, WD40 proteins interact with Aux/IAA proteins and co-repressors of TOPLESS (TPL) or TPL-RELATED (TPR) proteins. These interactions involve the C-terminal double WD40 motifs of TPLs/TPRs, which recruit histone deacetylase (HDAC) complexes and chromatin-modifying proteins. This allows AUX/IAA to mediate nearby chromatin condensation and transcriptional suppression [35]. During seed germination, WDRs participate in the CUL4–DDB1 E3 ubiquitin ligase complex for the degradation of DELLA proteins, which are negative regulators of gibberellin signaling, with the ABI5 protein, a positive regulator of abscisic acid signaling, controlling the balance between seed dormancy and germination [36,37]. WDRs also modulate the response of plants to various biotic and abiotic stresses, such as drought, salinity, heat, cold, and pathogens. They do so by regulating the expression or stability of stress-related genes, such as those encoding heat shock proteins, osmotic stress proteins, and pathogenesis-related proteins [38,39]. The progression through the cell cycle is another function involving WDRs through SCF-type Cullin Ring E3 Ubiquitin Ligases (CRLs), which results in the degradation of cycle-dependent kinase inhibitors (CKIs) at the G1 to S phase checkpoint [20]. DWD-type E3 CRL complexes containing WDRs as subunits may also degrade positive regulators of the cell cycle progression machinery [17] or control the endoreplication cycles of trichomes [40].

Flowering is an important developmental process that involves sophisticated molecular mechanisms. Concerning flowering time control, numerous gene products and signaling pathways have been identified, and their roles have been extensively studied. The network of genes configures several flowering pathways, such as the autonomous, photoperiod, and vernalization pathways [41]. The photoperiod pathway regulates *CONSTANCE* (*CO*) expression and consequently the expression of *FLOWERING LOCUS T* (*FT*), which constitutes the moving signal from the leaves to a SAM that promotes flowering [42]. On the other hand, the autonomous and vernalization pathways target *FLC*, which is under significant and meticulous regulation at the post-transcriptomic and epigenetic levels [43]. All these pathways converge to regulate the expression of the floral integrators *FT* and/or *SUPPRESSOR OF OVEREXPRESSION OF CONSTANS 1* (*SOC1*), which in turn activate the floral meristem identity genes *LEAFY* (*LF*) and *APETALA1* (*AP1*) to initiate the transition of a SAM to a floral meristem [44,45]. However, new components constantly emerge and expand our knowledge regarding the mechanisms of anthesis. One of the master regulators in flowering time control is FLC, which is considered the key floral repressor of FT during vegetative growth. *FLC* expression, and therefore flowering time, is under significant and meticulous regulation at the epigenetic, post-transcriptional, and hormonal (gibberellic acid; GA; and brassinosteroid; BR) levels [46,47,48]. More specifically, DELLAs physically interact with FLC in a large complex that enhances the transcriptional repression of FT by FLC, resulting in delayed flowering. In the presence of GA, DELLAs are rapidly degraded by the 26S proteasome, and the inhibitory effect of FLC on FT transcriptional repression is eliminated [49]. Previous studies have shown that the presence of BRs activates FLC transcription through downstream signaling components such as the BZR1 and BES1-INTERACTING MYC-LIKEs (BIMs) transcription factors. BZR1 transcriptional activity is promoted directly through BR-mediated phosphorylation mechanisms and indirectly by the activation of GA biosynthesis, which results in elevated GA levels and the degradation of DELLAs, which results in the release of the R-activated BZR1 transcription factor. Moreover, BRI1-EMS-SUPRESSOR 1 (BES1) can recruit EARLY FLOWERING 6 (ELF6) and RELATIVE OF EARLY FLOWERING 6 (REF6) to regulate the target gene expression, such as FLC [50]. Both ELF6 and REF6 are involved in the histone modification mechanisms that control flowering time; ELF6 functions as a repressor of FT, while REF6 acts as a repressor of FLC [51,52]. Moreover, it has also been reported that Arabidopsis HSP90-impaired plants failed to promote a transition from the vegetative into the reproductive phase and showed flower morphological heterogeneity. The authors propose that HSP90 consolidates a molecular scaffold able to arrange and organize the flowering gene network during vegetative-to-reproductive phase transitions and to progress flowering [53]. HSP90 acts as a master regulator of the BR hormonal pathway through physical interactions with key signaling components such as the BRI1 and BAK1 receptors [54], the BIN2 kinase [55], and the BES1 and BZR1 transcription factors [56]. In addition, HSP90 regulates the crosstalk of the BR and GA signaling pathways since HSP90 interacts with DELLA proteins, such as RGA and GAI, and HSP90 activity is required for DELLA GA-promoted degradation and the expression of BZR1-dependent transcriptional targets [56].

Our results provide, for the first time, evidence of the existence of a mechanism that links flowering time control with the epigenetic regulation of *FLC* through the involvement of heat shock proteins. We have previously shown that APRF1 and ULCS1 are two closely related WDRs from Arabidopsis. Although they seem to have unique roles in plant development, their unrecognized functions may well be more multi-dimensional. Both proteins are involved in the formation of multifunctional protein complexes that regulate key processes in plants [16,17,18]. More specifically, APRF1 appears to play a central role in flowering time control through the regulation of *FLC* at the epigenetic level. In *aprf1* mutants, the deregulation of *FLC* expression results in a delay in plants entering their reproductive phase [18]. Here, we show that the late-flowering phenotype of *aprf1* can be obviated at higher growth temperatures. When plants were grown in elevated temperature conditions, both the *aprf1* mutants and WT plants flowered at similar times, indicating that the delayed flowering of *aprf1* could be complemented by a mechanism involving the function of heat shock proteins. To further investigate this phenomenon and gain a deeper insight into the components involved in the regulation of flowering time, we applied a Co-IP approach using as bait an APRF1-FLAG-tagged chimeric protein. The results of the pulldown experiment showed that APRF1 interacts at a high level of confidence with numerous nuclear and cytoplasmic proteins involved in multiple biological pathways and cellular processes. These include proteins that function in photosynthesis, proteolysis, responses to abiotic stimuli, binding to other proteins, DNA, and RNA, as well as in RNA regulation, processing, and translational initiation. Of note, the latter is in line with data demonstrating the existence of a co-transcriptional RNA process that modulates a chromatin regulatory mechanism controlling the quantitative expression of the *FLC* locus [57]. Furthermore, a recent report from the same laboratory showed that a phosphatase module of the mRNA 3′ processing CPSF complex (Cleavage and Polyadenylation Specificity Factor) is central to this mechanism. The function of the CPSF complex is essential for chromatin modification at the *FLC* locus. It results in the removal of H2K4 monomethylation in the central section of the *FLC* locus and, consequently, its transcriptional silencing. The group found that this phosphatase module functions genetically downstream of the CPSF complex and consists of the components TOPP4 (Glc7/PP1), LD (LUMINIDEPENDENS/Ref2/PNUTS), and APRF1. Thus, the authors conclude that the physical association of the CPSF phosphatase module, which includes the APRF1 protein, with chromatin modifiers generates a transcription-coupled silencing mechanism [58]. Moreover, APRF1 has also been identified in a previous study as a component of the so-called autonomous pathway complex (AuPC) [59]. The authors reported that the AuPC is a multi-subunit complex that contains the known components FLD, LD, and SDG26, as well as the newly characterized components EFL2, EFL4, and APRF1, which interact with FLD. Mutations in all of these subunits results in H3Ac, H3K4me3, H3K36me3 and H3K27me3 histone modifications, leading to increased *FLC* expression and delayed flowering. Thus, the authors conclude that the AuPC complex is involved in the regu-lation of multiple histone modifications at *FLC* chromatin [59]. 

Out of the list of immunoprecipitated proteins, we also identified a “protein folding chaperone” category, with several members belonging to the HSP70 and HSP90 protein families. In view of the physiological data regarding the temperature-dependent phenotype of the *aprf1* mutants and the report by Margaritopoulou et al. [53], showing that HSP90 organizes a molecular scaffold mechanism to progress flowering, we first aimed at testing whether the APRF1 and HSP90 proteins meet the requirements to form complexes in silico. By using AlphaFold2, we show that HSP90.1 and HSP90.2 members could be complexed with high confidence with APRF1. Both predictions exhibited a high number of putative salt bridges, hydrogen bonds, and non-bonded contacts, indicating the ability of APRF1 to interact with both HSP90.1 and HSP90.2. We therefore proceeded to verify whether this interaction also occurs in planta. By using a BiFC approach, we initially tested if APRF1 interacts with the constitutively expressed protein HSP90.2. The strong positive signal obtained in the cytoplasm of the cells verified that the two proteins do indeed interact in planta. To further investigate if other members of the HSP90 family may form complexes with APRF1, we also tested, to this end, the heat-inducible protein HSP90.1. Our data confirmed that HSP90.1 also interacts with APRF1, although in a different subcellular compartment. In the latter experiment, the interaction occurred exclusively in the nucleus of the transformed cells, suggesting the involvement of the HSP90.1/APRF1 complex in a different molecular mechanism compared to that of HSP90.2/APRF1.

Based on previous reports [18,58,59] and the data given here, a simplified hypothetical model can be depicted to illustrate the involvement of APRF1 and HSP90.1 during anthesis (Figure 7). According to this, flowering time is controlled at the epigenetic level by a complex that regulates *FLC* expression. During the transition of plants from the vegetative to the reproductive phase, the complex consisting of the components SDG26, FLD, LD, APRF1, and HSP90 suppresses *FLC* expression, thus allowing SOC1 to initiate anthesis (Figure 7a). In the absence of APRF1 [18,58] or APRF1 and SDG26 [59], flowering is delayed due to high levels of *FLC* expression (Figure 7b,c). Nevertheless, when plants are grown at elevated temperatures, the expression of the heat-inducible protein *HSP90.1* probably stabilizes the complex even in the absence of APRF1, thus allowing *aprf1* plants to enter their reproductive phase in the correct time frame (Figure 7d). Taken together, our results strongly support the dual localization of the APRF1 protein and indicate that its placement in different subcellular compartments may lead to the assembly of diverse functional complexes. As far as flowering time control is concerned, our data indicate that the anthesis-promoting function of APRF1 is exerted at the level of the transcriptional regulation of *FLC* through a complex that also involves a member of the HSP90 protein family. Since APRF1 has been shown to participate in the CPSF/AuP complex, it is reasonable to speculate that HSP90.1 may also be a member of this complex that balances its function according to the growth temperature. Clearly, further research is needed to identify additional components of this regulatory complex and to fully dissect and understand this flowering time framework.

## 4. Materials and Methods

### 4.1. Plants Growth and Treatments

For the present study, *Nicotiana benthamiana*, *Arabidopsis thaliana* (L.) Heynh. (ecotype Col-0), and APRF1 T-DNA insertion lines (WiscDsLox345-348I5 and WiscDsLox489-492K11) were used. The insertion lines were obtained from NASC (Nottingham, UK) and analyzed as described previously [18]. The seeds were initially imbibed at 4 °C for 48 h. For the soil experiments, the seeds were directly sown on the soil and transferred into a growth chamber. For the Petri-dish experiment, the seeds were surface-sterilized with 70% ethanol and 0.01% Triton X-100 for 10 min, then washed twice in 100% ethanol for 30 s, and finally spread on plant growth medium containing half-strength Murashige and Skoog Basal Medium with Gamborg′s vitamins (Sigma, Kawasaki, Japan, M0404), 0.5 g/L of MES (Sigma, E169) at pH 5.8 (calibrated with KOH), and 9 g of agar (Sigma, A1296). All seeds were germinated and grown at 21 °C in a 60–70% relative humidity with a light regime of 16 h of light and 8 h of darkness (long days, LDs) and an illumination of 110 μmol m^−2^ s^−1^ PAR, supplied by cool white fluorescent tungsten tubes (Osram, Munich, Germany). For the short-day (SD) conditions, plants were grown with a light regime of 8 h of light and 16 h of darkness. For protein isolation, green parts of two week-old soil-grown plants were harvested and placed in falcon tubes containing 10–15 steal beads and directly frozen in liquid nitrogen. For the heat shock treatments, the seeds sown directly on the soil were transferred to a growth chamber at 26 °C under LD settings.

### 4.2. PCR, Generation of Binary Vectors, BiFC, and Plant Transformation

Total RNA was extracted from the plant tissues with NucleoSpin^®^ RNA Plant kits, according to the manufacturer’s instructions (Macherey Nagel, Düren, Germany), and quantified using a NanoDrop 1000 spectrophotometer (Thermo Scientific, Waltham, MA, USA). Then, 1 μg of total RNA was used as a template for first-strand cDNA synthesis using PrimeScript Reverse Transcriptase (Takara-Clontech, Shiga, Japan) or Expand Reverse Transcriptase (Roche Diagnostics Ltd., Mannheim, Germany). To generate the appropriate constructs by cloning, the PCR products were amplified with Phusion^®^ High-Fidelity DNA Polymerase (New England Biolabs, Ipswich, MA, USA) or with ExpandTM High-Fidelity DNA Polymerase (Roche, Mannheim, Germany). Taq DNA polymerase (Invitrogen, Carlsbad, CA, USA) was used for semi-quantitative RT-PCR, according to the manufacturer’s instructions. For RNA calibration, transcripts of *GAPDH* were monitored as an internal control in all PCR reactions, which were performed in triplicate. For visualization, the PCR products were separated by electrophoresis on 0.9% agarose gels and photographed under UV light after staining with 100 μg L^−1^ of ethidium bromide.

For the subcellular localization of APRF1, a YFP::APRF1 translational fusion construct was generated as described previously [18]. To generate the APRF1 FLAG-tagged construct containing the N-terminus FLAG epitope, the cDNA coding sequence of *APRF1* lacking the translation stop codon was initially amplified using the primers APRF1-FLAG-F, containing an XbaI restriction site, and APRF1-FLAG-R, containing a SacI restriction site. The PCR product was digested, separated by electrophoresis on 1% agarose gel, and purified by using the NucleoSpin Gel Extraction and PCR Clean-up kits (Macherey Nagel, Germany). Subsequently, the fragment was ligated into the respective sites of the linearized pBI121 binary vector, resulting in plasmid p35S::cAPRF1::FLAG. The generated construct was checked for integrity and cloning correctness by a restriction enzyme analysis and DNA sequencing. The plasmid was then introduced into *Agrobacterium tumefaciens* GV3101 competent cells, and a confirmed colony was used for *Nicotiana benthamiana* leaf infiltration experiments to verify the expression of the recombinant protein. The protein samples were separated by SDS-PAGE and transferred onto nitrocellulose membranes, and the FLAG-tagged APRF1 fusion protein was detected with a primary anti-FLAG antibody (sc-807) and a secondary goat anti-rabbit IgG antibody (sc-2004) (Santa Cruz Biotechnology, Dallas, TX, USA). Protein bands were visualized by using the LumiSensorTM Chemiluminescent HRP Substrate Kit (L00221V300, GenScript).

A bimolecular fluorescence complementation (BiFC) protocol was performed according to Walter et al. [60] using the pSPYNE/pSPYCE vector system. The HSP90.1–YFPc and HSP90.2–YFPc constructs were generated as described previously [54]. The *APRF1* ORF was amplified by PCR from Col-0 cDNA using the primer pair *APRF1*–Xba-SpyNE and APRF1–Xho-SpyNE. The APRF1–YFPn construct was generated by cloning the purified *APRF1* band as an Xba-Xho fragment into the corresponding sites of the pSPYNE binary vector. All the constructs were introduced into *Agrobacterium tumefaciens* cells (strain GV3101) by using the freeze–thaw method. Bacterial colonies were selected and used to infiltrate the leaves of 4–6-week-old *Nicotiana benthamiana* plants. Leaf epidermal samples were observed after 48–72 h using an Axioscope fluorescence microscope. pSPYNE-35S::bZIP63 and pSPYCE-35S::bZIP63 were used as positive controls. As negative controls, the APRF1–YFPn construct was used along with the empty YFPc vector, and the HSP90 constructs (HSP90.1–YFPc or HSP90.2–YFPc) were used along with the empty YFPn vector. The primers used in this study are listed in Appendix A.

### 4.3. Protein Extraction

Ten grams of plant tissues from each line were collected and ground into powder with a vertical tissue homogenizer (1600 MiniG^®^, SPEX SamplePrep) while frozen in liquid nitrogen. The powder was resuspended into 20 mL of lysis buffer, containing 10% glycerol, 25 mM of Tris at pH 7.6, 1 mM of EDTA, 150 mM of NaCl, one protease inhibitor tablet (Thermo Fisher, Waltham, MA, USA, A32965), and 0.1% NP40 (Thermo Fisher, FNN0021). The mixtures were incubated in a rotator at 4 °C for 20 min. Next, the samples were centrifuged at 5500× *g* for 30 min at 4 °C. The debris from the supernatants was filtered through a double-layer miracloth filter (Merck, Rahway, NJ, USA, 475855), resulting in the total lysate. In order to concentrate the total lysate (about 20 mL), concentration columns (pore size: 10K, Pall, MAP010C38) were used. The final volume of each plant’s total lysate was reduced to 2 mL.

### 4.4. Protein Co-Immunoprecipitation

Magnetic dynabeads coated with protein G (Thermo Fisher, 10004D) were incubated with a monoclonal anti-FLAG antibody (Merck, F3165) for 60 min, rotating at 4 °C. For each sample, 60 μL of beads were incubated with 6 μg of the respective antibody. After washing with lysis buffer, the resulting complex was mixed with 1 mL of the concentrated total lysate. The mixture was placed into a rotator for overnight incubation at 4 °C. After three washes with lysis buffer, the beads were resuspended in the elution buffer, consisting of 45% PBS, 45% NuPAGE^®^ LDS Sample Buffer (Thermo Fisher, NP0007), and 10% beta-mercaptoethanol. The mixture was heated up to 90 °C for 10 min and, next, placed in a magnetic rack. The resulting supernatant containing the immunoprecipitation products was finally eluted from the beads.

### 4.5. Western Blotting

The protein samples were mixed with NuPAGE^®^ LDS Sample Buffer in the recommended ratios and resolved on Criterion TGX Precast Protein Gels of 4–20% poly-acrylamide gel (BioRad, Hercules, CA, USA, 5671094), then transferred onto a PVDF membrane with a 45 μm pore size (Merck, Rahway, NJ, USA) with the “sandwich” method. Then, the membrane was blocked for 6 h using 5% (*w*/*v*) bovine serum albumin (Sigma-Aldrich, A9647) dissolved in 1× Tris-buffered saline with Tween-20 (TBS-T) (Cell Signaling Technology, Danvers, MA, USA, 9997) under shaking at 4 °C. Next, a monoclonal anti-FLAG antibody (Merck, Rahway, NJ, USA) was diluted in TBS-T at a final concentration of 1 μg/mL and incubated overnight with the blotted membrane. After three washes with TBS-T, the membrane was incubated with the secondary antibody (Anti-mouse IgG-HRP, Cell Signaling Technology, 7076). The secondary antibody was used at a concentration of 0.1 μg/mL, and the incubation lasted for 45 min, shaking at room temperature. After three washes with TBS-T, the membrane was imaged with the ChemiDoc MP Imaging System (Bio-Rad, Dubai, United Arab Emirates) using the ECL Prime Western Blotting System (Merck, GERPN2232) as a substrate.

### 4.6. Mass Spectrometry

The gel-assisted proteomic sample generation approach was performed as previously described by Fischer and Kessler, 2015 [61]. Briefly, the samples were collected after the elution step and loaded on Criterion TGX Precast Protein Gels of 4–20% poly-acrylamide gel (BioRad, Dubai, United Arab Emirates). After this, the gel was stained with Coomassie Brilliant Blue R-250 Staining Solution (BioRad, 1610436) and then destained using the Coomassie Brilliant Blue R-250 Destaining Solution (BioRad, 1610438). The parts of the gel containing the proteins were removed, cut into small pieces, and placed into another destaining buffer (200 mM of Triethylammonium Bicarbonate (TEAB) (Thermo Fisher, 90114), 40% acetonitrile (ACN)). The mixtures were incubated at 37 °C under a shaking setting until the color of the gel pieces was totally removed. Subsequently, the supernatant was removed, and the gel pieces were dried using Concentrator Plus (Eppendorf, GmbH, Hamburg, Germany) and then resuspended in 100 mM of TEAB. In each sample, 0.3 μg of Sequencing Grade Modified Trypsin (Promega, Madison, WI, USA, V5111) was added. Overnight digestion with shaking at 37 °C was implemented. The supernatant containing the digested peptides was transferred into a new tube to undergo desalting using C18 pipette tips (Agilent, Santa Clara, CA, USA, 5188-5239). The resulting peptides were resuspended in 3% ACN and run into a Q-Exactive HF mass spectrometer (Thermo Fisher, Dreieich, Germany) coupled with an UltiMate™ 3000 UHPLC (Thermo Fisher, Dreieich, Germany). The raw data were aligned to the TAIR *Arabidopsis thaliana* database via Mascot 2.8 software.

### 4.7. Microscopy

All samples were visualized microscopically by using a Zeiss Axioscope (Zeiss, Oberkochen, Germany). The epifluorescence microscope was equipped with a differential interference contrast (DIC) system and proper filter sets: a filter set with exciter BP450–490 and barrier BP515–595, a set with exciter G-365 and barrier LP420, and a set with exciter BP510–560 and barrier LP590. Images of the samples were taken with a Zeiss Axiocam MRc5 digital camera. A Zeiss Stemi 2000-C stereomicroscope, equipped with a Jenoptic ProGres3 digital camera, was used for sample observation (Jenoptik, Jena, Germany). Microscopical images were processed using Adobe Photoshop CC Extended software (Adobe Systems Inc., San Jose, CA, USA), applying only basic adjustments.

### 4.8. Functional Enrichment Analysis and Assessment of Protein–Protein Interactions via STRING

The functional enrichment analysis was performed using the specialized bioinformatics platform FLAME [21], (https://bib.fleming.gr:8084/app/flame, accessed on 11 December 2023), which calculated the enrichment percentage of “Gene Ontology” (GO) terms and KEGG pathways derived from the 471 non-redundant putative interacting proteins of our study compared to a set of background proteins. The GO terms represent the “Biological Functions” (BF) of proteins, which are categorized into three main levels: “Molecular Function” (MF), “Biological Process” (BP), and “Cellular Component” (CC). The FLAME query variables selected for this analysis were the following: “Organism: *Arabidopsis thaliana*” (NCBI Tax. ID: 3702), “Enrichment tool: gProfiler”, “Background: 23,212 genes”, “Data sources: GO:MF, GO:CC, GO:BP and KEGG“, “Significance metric: false discovery rate”, and “Significance threshold: 0.05”.

STRING (Search Tool for the Retrieval of Interacting Genes/Proteins) is a database of known and predicted protein interactions in the form of direct (physical) or indirect (functional) associations, aiming to provide a critical assessment and integration of protein–protein interactions [23]. The settings used in Figure 5a were defined as “Network” type: full STRING network, meaning of network edges: confidence, active interaction sources: “Experiments, Databases, Co-expression, Neighborhood, Gene Fusion and Co-occurrence”, minimum required interaction score: high confidence 0.700, and hiding disconnected nodes in the “Network” (https://string-db.org/, accessed on 19 December 2023). For Figure 5b, the settings applied were defined as “Network” type: full STRING network, meaning of network edges: confidence, active interaction sources: all, and minimum required interaction score: medium confidence 0.400 (https://string-db.org/, accessed on 19 December 2023).

### 4.9. Alphafold2 Structures and Molecular Docking

For the protein–protein docking analysis, the AlphaFold2 [62,63] models of APRF1 (AF-Q9LYK6-F1), HSP90.1 (AF-P27323-F1), and HSP90.2 (AF-P55737-F1) were acquired from the neural-network-based AlphaFold2 Protein Structure Database (https://alphafold.ebi.ac.uk, accessed on 6 December 2023), an extensive database of high-accuracy protein structure predictions. The docking of APRF1-HSP90.1 and APRF1-HSP90.2 was conducted via the automated protein docking server ClusPro 2.0 [25,26], using the default settings. The generated clusters were evaluated using the balanced scoring scheme. Protein–protein interface statistics were evaluated through the “Generate” option in the EMBL-EBI PDBsum server [29,30].

## Figures and Tables

**Figure 1 ijms-25-01313-f001:**
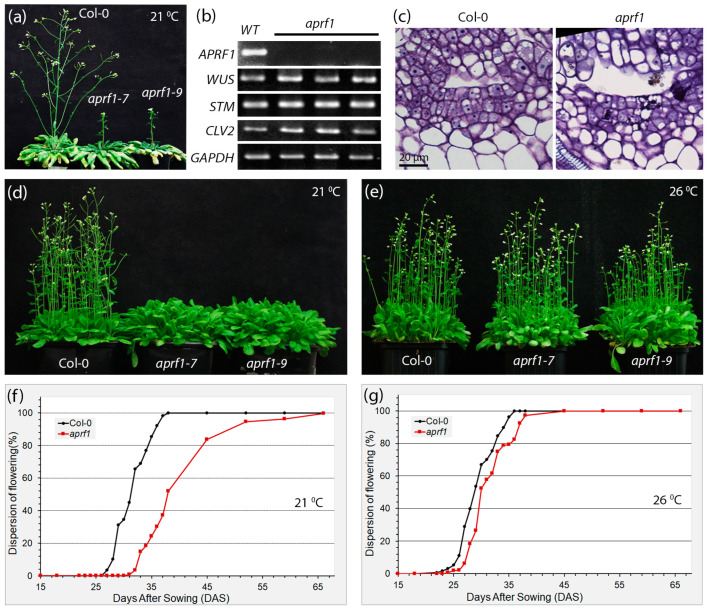
Thermoregulated late-flowering phenotype of *aprf1*. (**a**) Phenotypes of WT, *aprf1-7*, and *aprf1-9* mutants after growing plants for two months under short-day (SD) conditions (photoperiod cycle of 8 h light/16 h dark) and then transferred to long-day (LD) conditions (photoperiod cycle of 16 h light/8 h dark) for 14 days. (**b**) Semi-quantitative RT-PCR analysis showing the expression of the *APRF1*, *WUS*, *STM*, and *CLV2* genes in meristematic tissues of WT and *aprf1* mutant plants 24 DAS. Expression of *GAPDH* was monitored as control. (**c**) Longitudinal sections showing the meristematic cells of the SAMs from a representative WT and mutant plant after toluidine blue-stained 7 DAS. (**d**) Phenotypes of WT (Col-0), *aprf1-7*, and *aprf1-9* plants grown under LD conditions for 34 days at 21 °C. (**e**) Col-0 and *aprf1* mutants showing a WT phenotype after growing plants under LD conditions for 32 days at 26 °C. (**f**) Flowering time curves of WT and *aprf1* mutants growing under LD conditions at 21 °C. (**g**) Flowering time curves of WT and *aprf1* mutants growing under LD conditions at 26 °C, showing the temperature-related complementation of the late-flowering phenotype (*n* = 500). DAS, days after sowing.

**Figure 2 ijms-25-01313-f002:**
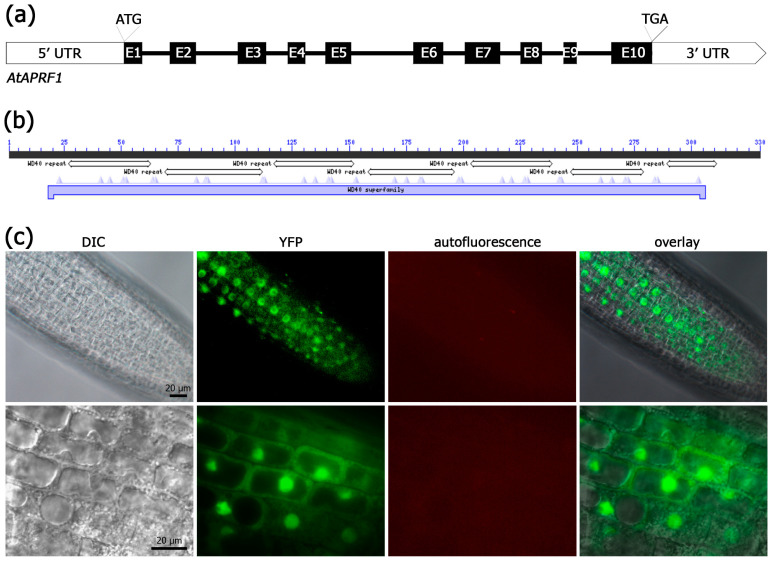
Genomic organization and subcellular localization of APRF1. (**a**) The *APRF1* gene consists of 10 exons (black boxes) and 9 introns (black lines). (**b**) The protein contains seven WD40 motifs, which generate a strong WD40 interacting domain spanning almost the entire protein sequence. (**c**) Images showing the localization of the YFP::APRF1 recombinant protein in Arabidopsis roots. A strong YFP fluorescence signal is observed in both the nuclei and the cytoplasm of the epidermal cells of the root meristematic zone. DIC, differential interference contrast images; YFP, images with GFP filters; red filter, autofluorescence of chlorophyll; overlay, YFP images superimposed on DIC images.

**Figure 3 ijms-25-01313-f003:**
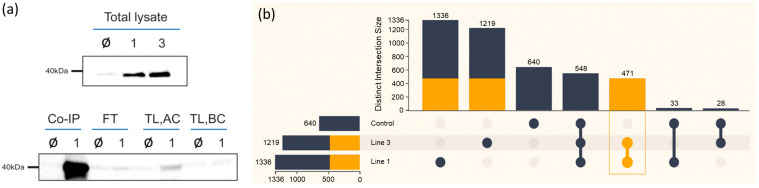
APRF1-FLAG expression and co-immunoprecipitation (Co-IP). (**a**) An equal amount of total lysate from the wild-type (WT) (Ø), line 1 (1), and line 3 (3) plants was blotted against the anti-FLAG antibody (**upper** panel). Equal volumes of total lysate (TL) in the before concentration (BC) and after concentration (AC), as well as the co-immunoprecipitation (Co-IP) product and the flow-through (FT) sample, were blotted against the anti-FLAG antibody to verify the successful immunoprecipitation of the APRF1-FLAG target protein (**lower** panel). (**b**) UpSet plot depicting the distinct intersections among the line 1, line 3 (orange color), and control Co-IP (Co-ImmunoPrecipitation) protein lists, as derived from the FLAME bioinformatics platform.

**Figure 4 ijms-25-01313-f004:**
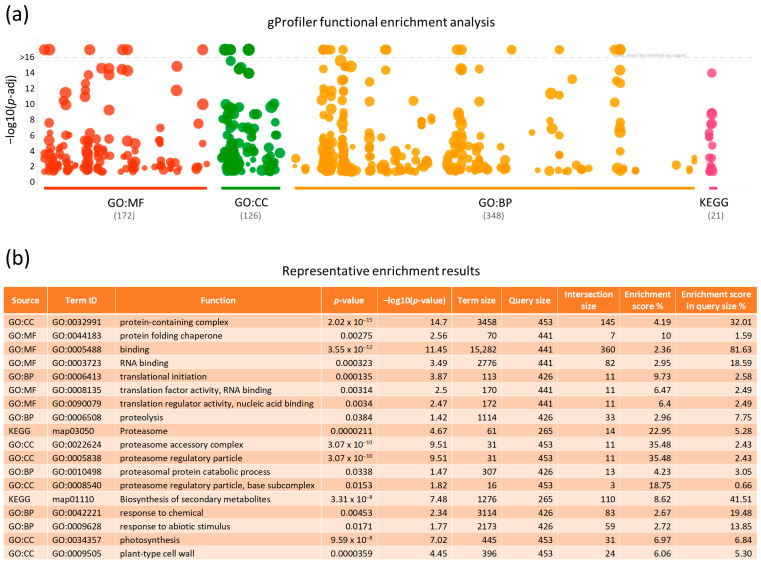
Functional enrichment analysis of the 471 non-redundant putative interacting proteins derived from APRF1 Co-IP (co-immunoprecipitation). (**a**) Manhattan plot describing on the *x*-axis the functional enrichment analysis of the 471 non-redundant proteins via the gProfiler pipeline in the categories “GO:MF” (Molecular Function; red color), “GO:CC” (Cellular Component; green color), “GO:BP” (Biological Process; orange color), and KEGG pathways (purple color), while the *y*-axis depicts the significance (*p*-value) of each term. The number of enriched functions per data source is presented in parentheses. (**b**) Table of representative functional enrichment results, combining different GO terms and KEGG pathways. For each term, the *p*-value and enrichment score (%) are described, as well as additional important features of the analysis. The full list of the entire functional enrichment analysis is provided in Appendix A.

**Figure 5 ijms-25-01313-f005:**
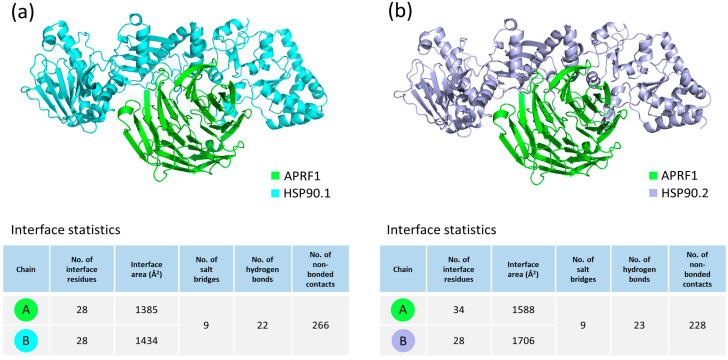
Molecular docking interactions between the APRF1 and HSP90 chaperone family members via the employment of the automated protein docking server ClusPro 2.0. Three-dimensional (3D) visualization, via the PyMOL molecular graphics system [28], of the protein–protein docking analysis of (**a**) APRF1 (green) and HSP90.1 (cyan) and (**b**) APRF1 (green) and HSP90.2 (magenta), AlphaFold2-derived structures. Interaction interface statistics of the docked complexes, generated by the EMBL-EBI PDBsum [29,30] web server, are also provided.

**Figure 6 ijms-25-01313-f006:**
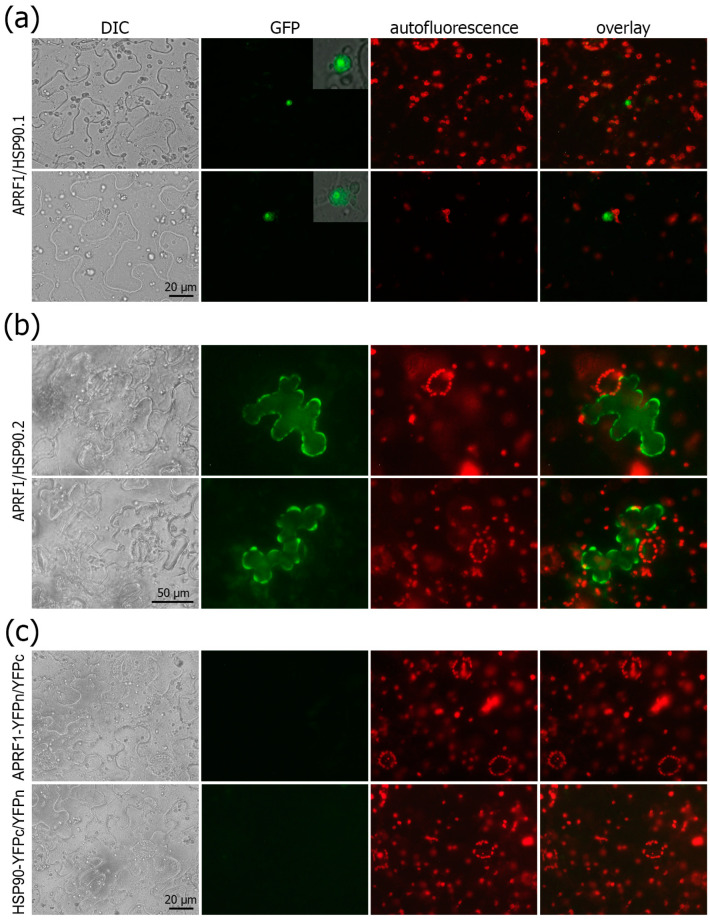
Subcellular localization and protein interaction of APRF1. (**a**) Representative images showing the specific in planta localization and interaction of the ARPF1 and HSP90.1 proteins in the nucleus of two epidermal cells (**upper** and **lower** panel). Insets show magnification of the nucleus, where the YFP signal is prominent in the nucleoplasm and the nucleolus. (**b**) Representative images illustrating the specific in planta localization and interaction of the ARPF1 and HSP90.2 proteins in the cytoplasm of two epidermal cells. (**c**) Representative control images showing that the co-transformation of the construct APRF1–YFPn along with the empty YFPc vector (**upper** panel) or HSP90.1/Hsp90.2 (HSP90) and the empty YFPn vector (**lower** panel) did not result in any fluorescence signal. Cells were visualized under an epifluorescence microscope equipped with DIC optics and GFP filter sets. No fluorescent signal was obtained with either of the three proteins used alone. DIC, differential interference contrast images; YFP, images with GFP filters; red filter, auto-fluorescence of chlorophyll; overlay, YFP images superimposed on DIC and auto-fluorescence images. Scale bars: 20 μm for (**a**) and 50 μm for (**b**).

**Figure 7 ijms-25-01313-f007:**
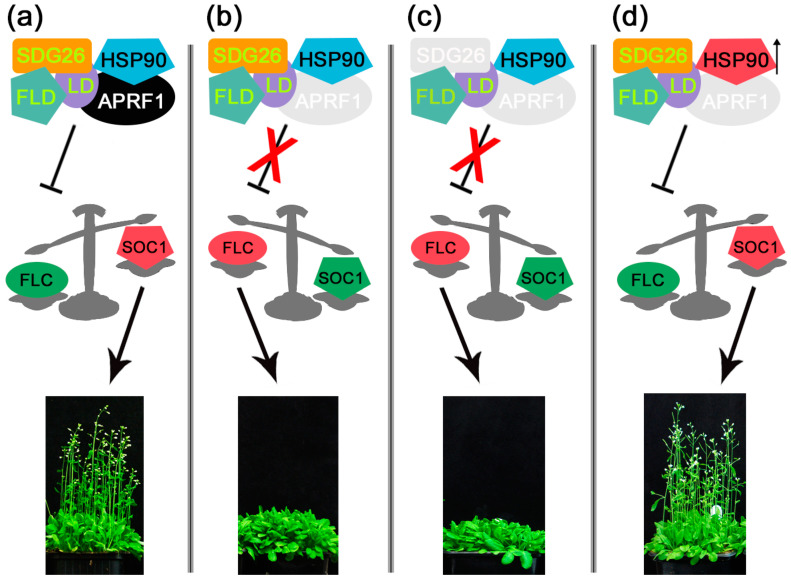
Simplified hypothetical model illustrating the involvement of APRF1 and HSP90.1 in anthesis. (**a**) The complex containing the subunits APRF1 and HSP90 suppresses FLC expression during anthesis, thus allowing SOC1 to initiate flowering. (**b**,**c**) The absence of the key components APRF1 and/or SDG26 results in delayed flowering due to the presence of high FLC protein levels. (**d**) At elevated growth temperatures, the expression of the heat-inducible gene *HSP90.1* (upwards pointing arrow) results in the maintenance of a functional complex, even in the absence of APRF1, thus leading to the successful anthesis of the plants in the correct time frame. SDG26, LD, and FLD, previously reported as members of the complex, are indicated with yellow letters [58,59]. Refer to the text for details.

## Data Availability

All the data supporting the findings of this study are available within the paper and in its Appendix A, published online. The plant materials used in this study are available upon request.

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
