# Peer review of "APRF1 Interactome Reveals HSP90 as a New Player in the Complex That Epigenetically Regulates Flowering Time in Arabidopsis thaliana"

_ijms, 2024, doi:10.3390/ijms25021313_

Round 1
Reviewer 1 Report
Comments and Suggestions for Authors
Comments on the Quality of English LanguageMinor editing of English language required.
Author Response
Response to Reviewer #1 for Manuscript ID: ijms-2824821 (Isaioglou et. al.)
The manuscript "Functional interaction of APRF1 and HSP90 promotes thermoregulated flowering of Arabidopsis thaliana under long days" presents lack of APRF1 delays plant flowering but can suppress the phenotype under mild temperature. Authors identified its interaction proteins by Co-IP and MS-SPEC. Further, APRF1 interacted with HSP90.1 and HSP90.2 physically. Overall, the authors provide useful information and represent a valuable contribution to an important subject. However, the manuscript can be significantly improved. The written presentation and readability can be improved by correcting typological errors, and well illustrating of some figures. The following comments are suggestions that are important to address to improve the overall quality of this manuscript.
Answer: We thank the Reviewer for the excellent review and his/her constructive criticism. The Reviewer brought up several points in the MS, which we agree with. The Reviewer helped us to correct several mistakes and oversights and to significantly improve the manuscript. We have addressed all comments and hope that the Reviewer is satisfied with the performed changes and additions. Detailed responses to each of the comments are as follows.
Major points:
- Data from some databases had better be as supplementary figs. For example, fig.5 and/or fig. are better as supplementary data.
Answer: We moved Figure 5 to Supplementary Data, now Supplementary Figure S1, and revised the text accordingly.
- Fig.7. should clearly present the parts of interactions in line (upper, bottom), and add negative control.
Answer: The upper and lower panels in (a) and (b) show representative images of two individual cells from repetitive experiments. We now made this clear in the legend. As suggested by the Reviewer, we also included in Figure 7 (now Figure 6) negative controls (as c).
Minor points:
- Line 106. Figure 1D; Fig. 1A-C should be referenced more early than 1D.
Answer: We have revised the text so that references of 1a-1g are continuous.
- 7 DAY. What kind of cells? Cotyledon? or 1st real leaf? Please clarify the legend.
Answer: Image 1c are longitudinal sections showing the meristematic cells of the shoot apical meristem (SAM) from a representative WT and mutant plant after toluidine blue-stained at 7 days after sowing. We have now made this clear in the legend.
- Line 154. Comma after (Fig.2a and 2b).
Answer: Dot was added after ….(Figure 2a and 2b).
- Line 162. [18] should be replaced by Fig. 2C.
Answer: In this line, we refer to [18] based on our previous report (Kapolas et al., 2016) where we showed the localization of APRF1 in more mature root cell types (differentiation zone). In these cells, the YFP signal was observed predominantly in the nucleus of the cells. YFP could also be seen in the cytoplasm, but the signal was quite faint. However, by repeating the experiment and carefully observing the meristematic zone a strong YFP signal can be seen both in the nucleus and the cytoplasm of the meristematic cells (this report).
- Line 186. A. thaliana should be at slash.
Answer: We have corrected Arabidopsis to “A. thaliana”.
- Line 288-308. It is more likely to describe the methods rather than interpret the result in Result 2.6.
Answer: We have revised and elaborated “Result section 2.6” in order conform with the comment of the Reviewer.
- Fig.8 Please improve the artistic illustration of the model proposed. Currently, the model is difficult to understand and does not accurately illustrate the discovered mechanism by which plants respond to temperature. For example, no data is about SDG26 in the manuscript.
Answer: We agree with the Reviewer and have revised Figure 8 (now Figure 7) to conform with his/her comment. We believe that the Figure is now clearer to read and correctly illustrates the mechanism. We also made clear in the legend that SDG26, LD and FLD are subunits of the complex previously identified by Qi et al., 2022 and Mateo-Bonmatí et al., 2023 (now with yellow letters in the Figure).

Reviewer 2 Report
Comments and Suggestions for Authors
In this manuscript, Isaioglou et al. have provided an examination of the APRF1 protein and its role in regulating temperature-dependent flowering in Arabidopsis. The manuscript is well-written and structured appropriately. However, the title appears to overstate the findings. To justify the current title, the authors should consider the following additions: 1. Conduct biochemical experiments to confirm the direct interaction between APRF1 and HSP90. 2. Identify and characterize the critical interaction residues between APRF1 and HSP90. 3. Perform mutational analysis of these interaction residues in APRF1, followed by a complementation assay in the aprf1 mutant, to determine if the restoration of the temperature-dependent flowering-time phenotype is specific to wild-type APRF1 and not the mutated variant. These experiments are crucial to substantiate the claim that the interaction between APRF1 and HSP90 facilitates temperature-dependent flowering in Arabidopsis. Otherwise, it is advisable to revise the title to avoid potential overstatement of the findings.
Specific Comments:
Line 21: Please include the full name for "WD40s." Similarly, provide full names for "aprf1" (Line 26, 29), "Co-IP" (Line 29), "HSP90" (Line 32), "COMPASS" (Line 50), and "CPF" (Line 51). Ensure thoroughness throughout the manuscript.
Line 64-65: Add a reference to support the claim regarding VIP's association with DDB1, and clarify the nature and implications of this association, if available.
Introduction: It's recommended to include a discussion on the regulation of flowering-time in Arabidopsis, as it is a central theme of the study.
Line 159: Clarify the discrepancy between the use of YFP-tagged APRF1 and the monitoring of GFP expression.
Line 163: Correct the figure reference to Figure 2C.
Figure 2C: Address the apparent inconsistency in the localization of APRF1 between the upper and bottom panels. The upper panels show YFP signals in both the nucleus and cytoplasm, whereas the bottom panels suggest localization to the nucleus and plasma membrane.
Line 179: "aprf1" should be italicized.
Figure 3a: Explain the faint band observed in wild-type plants when probed with FLAG antibody. Confirm if the bands in lanes 1 and 3 are indeed the correct APRF1-FLAG bands. Similar clarification is needed for the lower panel.
Line 215: It's unnecessary to provide the full name of "MS" here.
Line 329 and Figure 7a: Include a nuclear marker, such as DAPI, to support the claim made. Additionally, clarify the differences observed between the upper and lower panels.
Author Response
Response to Reviewer #2 for Manuscript ID: ijms-2824821 (Isaioglou et. al.)
We thank the Reviewer for the excellent review and his/her constructive criticism. The Reviewer brought up several points in the MS, which we agree with. The Reviewer helped us to correct several mistakes and oversights and to significantly improve the manuscript. We have addressed all comments and hope that the Reviewer is satisfied with the performed changes and additions. Detailed responses to each of the comments are as follows.
In this manuscript, Isaioglou et al. have provided an examination of the APRF1 protein and its role in regulating temperature-dependent flowering in Arabidopsis. The manuscript is well-written and structured appropriately. However, the title appears to overstate the findings. To justify the current title, the authors should consider the following additions: 1. Conduct biochemical experiments to confirm the direct interaction between APRF1 and HSP90. 2. Identify and characterize the critical interaction residues between APRF1 and HSP90. 3. Perform mutational analysis of these interaction residues in APRF1, followed by a complementation assay in the aprf1 mutant, to determine if the restoration of the temperature-dependent flowering-time phenotype is specific to wild-type APRF1 and not the mutated variant. These experiments are crucial to substantiate the claim that the interaction between APRF1 and HSP90 facilitates temperature-dependent flowering in Arabidopsis. Otherwise, it is advisable to revise the title to avoid potential overstatement of the findings.
Answer: We agree with the Reviewer and have changed the title of the MS to “APRF1 interactome reveals HSP90 as a new player in the complex that epigenetically regulates flowering time in Arabidopsis thaliana”.
Specific Comments:
Line 21: Please include the full name for "WD40s." Similarly, provide full names for "aprf1" (Line 26, 29), "Co-IP" (Line 29), "HSP90" (Line 32), "COMPASS" (Line 50), and "CPF" (Line 51). Ensure thoroughness throughout the manuscript.
Answer: “WD40s” was replaced with “WD40-repeat proteins (WDRs)”. “aprf1” was changed to “anthesis promoting factor 1 (aprf1)”. "Co-IP" was changed to “Co Immunoprecipitation (Co-IP)”. "HSP90" was replaced with “HEAT SHOCK PROTEIN 1 (HSP90.1)”. "COMPASS" was changed to “COMPASS (Complex Proteins Associated with Set1)” and "CPF" was changed to “CPF (Cleavage and Polyadenylation Factor)”.
Line 64-65: Add a reference to support the claim regarding VIP's association with DDB1, and clarify the nature and implications of this association, if available.
Answer: Reference supporting the association of VIP3 with DDB1 was added. The nature and implications of this association is reported by the sentence in the text that follows: …“Together they are subunits of a putative E3 DWD CRL complex that negatively regulates flowering time in A. thaliana through ubiquitin-mediated degradation of specific target proteins [11,12]…”
It's recommended to include a discussion on the regulation of flowering-time in Arabidopsis, as it is a central theme of the study.
Answer: As recommended by the Reviewer, we have revised and elaborated the “Discussion” section concerning flowering time control in Arabidopsis. The section now includes adequate information regarding the flowering pathways and the network of the key genes involved in anthesis.
Line 159: Clarify the discrepancy between the use of YFP-tagged APRF1 and the monitoring of GFP expression.
Answer: GFP and YFP have similar emission peaks (509 and 527, respectively). Our microscope is equipped with a broadband filter that allows unobstructed observation of the YFP signal in the “Green” wavelength. This is common practice also by other researchers or protocols. However, we agree with the Reviewer that this is confusing and misleading. We have therefore corrected “GFP” to “YFP” in the text. Nevertheless, we state in the legend of Figure 2 and Figure 6 that the “YFP” columns show images obtained with GFP filters.
Line 163: Correct the figure reference to Figure 2C.
Answer: We have corrected reference to Figure 2c. We apologize for the mistake.
Figure 2C: Address the apparent inconsistency in the localization of APRF1 between the upper and bottom panels. The upper panels show YFP signals in both the nucleus and cytoplasm, whereas the bottom panels suggest localization to the nucleus and plasma membrane.
Answer: Please allow us to disagree with the Reviewer on this occasion. Both upper and lower panels show that APRF1 is localized to the nucleus and the cytoplasm. At higher magnifications, we observe a clear fluorescence signal at the periphery of the cells, which corresponds to the cytoplasm. In all epidermal cells the presence of a huge vacuole, which occupies almost the entire cell, displaces the cytoplasm to the periphery of the cells. The width of the YFP signal in the periphery of the cells is too wide to correspond to the plasma membrane.
Line 179: "aprf1" should be italicized.
Answer: All “aprf1” were italicized in the text.
Figure 3a: Explain the faint band observed in wild-type plants when probed with FLAG antibody. Confirm if the bands in lanes 1 and 3 are indeed the correct APRF1-FLAG bands. Similar clarification is needed for the lower panel.
Answer: We acknowledge that the faint band in the lysate of the wild-type plants may be a matter of concern. However, an equal amount of total lysate was loaded during the Western blot of Figure 3a from each sample. The faint band of Col-0 corresponds probably to an unspecific band. That's why there is an enormous difference between the intercity of the signal between Col-0 and lines 1 and 3. The confirmation that the bands from lines 1 and 3 are indeed APRF1-FLAG is the fact that in the Co-IP MS assay (lower panel) the APRF1-FLAG was identified in the proteome of lines 1 and 3 only, and not in the Col-0 sample. We suspect that the anti-FLAG antibody recognizes an unknown endogenous protein of Arabidopsis thaliana, only in its denatured/linearized form. We can conclude that, since this faint band can be observed in the lower panel Fig 3a in the Col-0 samples of Total Lysate, After Concentration (TL,AC) and Flow Through (FT) but not in the Co-IP sample. If the anti‑FLAG could recognize the epitope of this unknown protein in its native form, there would be a signal in the Co-IP sample of the Col-0, which would also be higher in intensity compared to the Col‑0 FT sample. Therefore, it is safe to conclude that the faint bands observed in Fig 3a result from nonspecific binding of the anti-FLAG antibody to the denatured form of an unknown protein. Nevertheless, and as the outcome of the experiment has proven, this has not affected, by no means, the quality of the Co-Immunoprecipitation.
Line 215: It's unnecessary to provide the full name of "MS" here.
Answer: “…Mass Spectrometry (MS) analysis.” was changed to “MS analysis”.
Line 329 and Figure 7a: Include a nuclear marker, such as DAPI, to support the claim made. Additionally, clarify the differences observed between the upper and lower panels.
Answer: We have revised Figure 7 (now Figure 6) according to the comments of Reviewer No1. We have now included negative controls to the Figure (as c). We have also added inserts of magnifications of the nucleus on the GFP images of panel (a). It is now clearly visible that the YFP signal corresponds to the nucleus. In fact, the signal appears to be stronger in the nucleolus than in the nucleoplasm. The upper and lower panels in (a) and (b) (and now in c as well) show representative images of two individual cells from repetitive experiments. We now made this clear in the legend.
